# Predictors of Fall-Related Injuries in Fallers—A Study in Persons with Cognitive Impairment

**DOI:** 10.3390/geriatrics10030074

**Published:** 2025-05-28

**Authors:** Per G. Farup, Knut Hestad, Knut Engedal

**Affiliations:** 1Department of Research and Innovation, Innlandet Hospital Trust, P.O. Box 104, N-2381 Brumunddal, Norway; knut.hestad@inn.no; 2Department of Clinical and Molecular Medicine, Faculty of Medicine and Health Sciences, Norwegian University of Science and Technology, N-7491 Trondheim, Norway; 3The Norwegian National Center for Aging and Health, Vestfold Hospital Trust, N-3103 Tønsberg, Norway; knutengedal@outlook.com; 4Department of Geriatric Medicine, Oslo University Hospital, N-0424 Oslo, Norway

**Keywords:** cognitive impairment, dementia, falls, fall-related injuries

## Abstract

Background/Objectives: Old age and cognitive impairment/dementia are risk factors for falling and fall-related injuries. We have, in a previous study in persons with cognitive impairment, shown that falls were associated with frailty, reduced physical fitness, and cognitive reduction. Falls were independent of the disorders causing the impaired functions. Because most falls are innocent, knowledge of predictors of fall-related injuries seems more clinically relevant than the predictors of falls. Predictors of falls and fall-related injuries are not necessarily identical. The aim of this follow-up study to our previous one in the same population was to explore predictors of fall-related injuries in fallers and compare these predictors with those of falls. Methods: This study and our previous study used data from the “The Norwegian Registry of Persons Assessed for Cognitive Symptoms” (NorCog), a Norwegian research and quality registry with a biobank. The registry included consecutive home-dwelling persons referred to Norwegian specialist healthcare units for assessment of cognitive decline. This study included 3774 persons from our previous study who experienced falls last year and compared persons with and without a fall-related injury. A fall-related injury was defined as admittance to a hospital for the injury. Results: The annual incidence of fall-related injuries in the fallers was 884/3774 (23.4%). Female sex, older age, lower BMI, in need of public health service and walking assistance, and low Hb and Ca were independent predictors of fall-related injuries, indicating reduced physical fitness and state of health and a high burden of comorbidity. Injuries were not associated with the degree of cognitive impairment or the dementia diagnosis. Conclusions: In home-dwelling persons with impaired cognitive functions and falls, fall-related injuries were associated with reduced physical fitness and state of health. In contrast to predictors of falls, neither the degree of cognitive impairment nor the dementia diagnosis was associated with fall-related injuries. The difference is comprehensible. Persons with cognitive impairment or dementia might have reduced power of judgment and be inattentive, unconcerned and careless, which increases the fall incidence but not the risk of injury once falling. Prevention of fall-related injuries should focus on relieving comorbidities, improving physical fitness and general health rather than on cognitive improvement.

## 1. Introduction

Old age and cognitive impairment are risk factors for falling [1,2,3,4]. Dementia and falls have a significant impact on quality of life and health expenses. In persons 75 years of age and older, falls are number eight on the list of causes of disability-adjusted life years, according to the Global Burden of Disease Study 2019 [5]. In Norway, dementia and falls were the conditions with the highest health spending [6]. We previously reported, in a study using data from The Norwegian Registry of Persons Assessed for Cognitive Symptoms (NorCog), an annual fall incidence of 3,774/9,525 (39.6%) in persons with impaired cognitive functions. Falls were associated with frailty, reduced physical fitness, and reduced cognitive performance. Falls were independent of the disorders causing frailty and impaired functions [7].

Fortunately, most falls are relatively innocent. Nearly half of all falls do not hurt, approximately 30–50% lead to minor injuries, like bruises and lacerations, and 5-10% result in more severe injuries, such as fractures and brain damage [3]. Predictors of falls and fall-related injuries are not necessarily identical. We are not aware of studies that compare risk factors for falls and fall-related injuries. Fall prevention is particularly important in persons at high risk of fall-related injuries.

The aim of this follow-up of our previous study was to explore predictors of fall-related injuries in fallers and compare these predictors with those of falls in our previous study in the same population with impaired cognitive function/dementia.

## 2. Materials and Methods

### 2.1. Data Material

The “The Norwegian Registry of Persons Assessed for Cognitive Symptoms” (NorCog) is a Norwegian research and quality registry with a biobank [8]. From 2009 to 2021, the registry included 18,120 home-dwelling persons referred to Norwegian outpatient specialist healthcare units for assessment of cognitive functions and dementia. All participants had the capacity to give informed consent. The data were obtained at the first visit to the clinic. Participation was voluntary, and written informed consent was given before inclusion. This study used data from NorCog.

### 2.2. Variables

The paper by Medbøen et al. gives an overview of all variables in the registry [8]. The information was provided by the participants and their next of kin. This study used the following variables:

#### 2.2.1. Person Characteristics

Sex (male/female); Age (years); Height (cm); Body weight (kg); Body Mass Index (BMI: kg/m^2^).Injured person: A person admitted to the hospital after a fall.Education (years in formal education).Living alone at home (yes/no).In need of regular public health service (yes/no).In need of walking assistance: no; indoor or outdoor; indoor and outdoor (scores 0–2).Physical activity was the sum of easy and strenuous activity (score 0–8).Gait speed: The time used for walking 4 m was measured and scored according to “Short physical performance battery” (score 0–4) [9,10]. A high score means fast walking.Balance test was measured according to the “Short physical performance battery” (results in seconds; normal > 10 s) [10,11].Basic Personal Activities of Daily Living (PADL) were measured using the “Physical Self-Maintenance Scale” (Score 6–30) [12,13]. High scores mean less self-reliance.Instrumental Activities of Daily Living (IADL) was measured with the “Lawton Instrumental Activities of Daily Living Scale (Score 8–31) [12,13]. High scores mean less self-reliance.Systolic and diastolic blood pressure (mmHg) and heart rate (beats/min) were measured while sitting and in the standing position for 1 and 3 min.

#### 2.2.2. Medical History, Cognitive Functions

Ten diseases, present or previous, were noted (yes/no): Cerebrovascular disease, Parkinson’s disease, Neurological disease, Coronary heart disease, Cardiac surgery, Diabetes, Cancer, Chronic obstructive pulmonary disease (COPD), Polyarthritis, and Depression (present).Cornell scale for measuring depression in dementia (Score 0–38. No depression < 7; mild depression 7–11; moderate–severe depression ≥ 12) [14].MAYO Sleep questionnaire. Question 8 of the questionnaire rates the person’s general level of alertness from sleeping to normally awake during the day (score 0–10) [15].MAYO composite fluctuation scale (Score 0–4. Dichotomized: normal 0–2 = 0; abnormal 3–4 = 1) [16]Duration of cognitive impairment assessed by the participant or next of kin (years).Mini-Mental State Examination—Norwegian third revised version (MMSE-NR3). Cognitive functions were evaluated using a slightly modified and validated Norwegian version of the international MMSE test, with a score of 0–30 [17,18]. Higher scores indicate better performance.CERAD word list. The test measures both immediate and delayed recall (after 10 min) of 10 words. The immediate test was the sum of three recalls. The word list was a demographically adjusted version of the Consortium to Establish a Registry for Alzheimer’s Disease (CERAD) [19,20,21]. Higher scores mean better performance.Trail Making Tests A and B (TMT-A and TMT-B). The tests measure mental flexibility (the results are given in seconds). A shorter time (lower scores) indicates better performance.Drugs (number of drugs taken regularly).

#### 2.2.3. Laboratory Tests

Blood tests: Hb (g/100 mL); CRP (mg/L); ESR (mm/1h); Thrombocytes (10^9^/L); Creatinine (µmol/L); Albumin (g/L); Folic acid (nmol/L); Cholesterol (mmol/L); Homocystein (µmol/L); Na (mmol/L); K (mmol/L); Ca (mmol/L); Thyroxin (pmol/L); TSH (mU/L); Vitamin B12 (pmol/L); Vitamin D (nmol/L); HbA1c (nmol/mol); Methyl Malonic Acid (MMA: µmol/L); ALAT (U/L); Gamma-GT (U/L); ALP (U/L).Cerebrospinal fluid: Phospho-tau (reference value < 80 pg/mL); Total-tau (reference values: age < 50: <300 pg/mL; age 50–70: <450 pg/mL; age > 70: <500 pg/mL); Beta-amyloid (reference value < 550 ng/L) [22].Genes: APOE genotypes: E2E2; E2E3; E2E4; E3E3: E3E4; E4E4.

#### 2.2.4. Clinical Evaluation and Diagnoses

All available information was evaluated, including the results of blood and cerebrospinal fluid tests, MRI or CT scans of the brain, and FDG-PET scans, and was discussed in interdisciplinary meetings before a final diagnosis was concluded.

Clinical evaluation: Subjective cognitive impairment (SCI); Mild cognitive impairment (MCI); Dementia. The diagnoses of SCI and MCI were defined according to Jessen et al. [23] and Winblad et al. [24], respectively.Dementia diagnoses: Alzheimer’s Disease (AD), Mixed AD/Vascular Dementia (AD/VaD), Vascular Dementia (VaD), Frontotemporal dementia (FTD), and Diffuse Dementia with Lewy Bodies (DLB). The diagnoses of FTD and DLB were according to Neary et al. [25] and McKeith et al. [26], respectively. The other diagnoses followed the ICD-10 criteria (the 10th revision of the International Statistical Classification of Diseases and Related Health Problems)

### 2.3. Statistics

Descriptive data are reported as numbers (proportion/percentage) and mean (SD). The unadjusted bivariate comparisons between fallers and non-fallers were analysed using a Chi-square test, with the trend if appropriate, and a *t*-test. The final comparisons between fallers and non-fallers were performed using a logistic regression analysis with the inclusion of age and sex, followed by a stepwise forward inclusion of other variables. Visual judgment of a Q–Q plot was used for the evaluation of the normality of the residuals. The goodness of fit was assessed with the Hosmer–Lemeshow test. The number of persons in each analysis is provided because of a relatively high proportion of missing data. Due to numerous analyses, *p*-values < 0.01 were considered statistically significant. The analyses were performed with IBM SPSS Statistics for Windows, version 29.0 (IBM Corp., Armonk, NY, USA).

## 3. Results

### 3.1. Person Characteristics

Of 18,120 persons in the registry, 9525 (53%) answered the question about falls. This study included 3774 (39.6%) persons who had a fall in the past 12 months, of whom 884 (23.4%) were injured and admitted to the hospital. Table 1 gives the person’s characteristics.

### 3.2. Comorbidity

Somatic and psychiatric comorbidity were compared between persons with and without a fall-related injury. Cerebrovascular disease, COPD and use of drugs were more prevalent in the injured. Table 2 gives detailed comparisons of previous and actual comorbidity, cognitive functions, and the actual use of regular drugs between injured and not injured.

### 3.3. Analyses of Blood and Cerebrospinal Fluid and Genetic Tests

The analyses of the blood and cerebrospinal fluid specimens and the genetic tests varied between fallers with and without an injury. Hb and Ca were significantly reduced, and ESR and homocysteine were elevated in the injured persons. Table 3 gives the details.

### 3.4. Clinical Evaluation and Final Diagnosis

The fall-related injuries were independent of the degree of cognitive impairment and dementia diagnoses. Table 4 gives the details with comparisons between persons with and without an injury.

### 3.5. Independent Predictors of Fall Injuries

Low BMI, need for public health service, and walking assistance were together with low serum-Ca independent predictors of fall injuries. Details are given in Table 5.

## 4. Discussion

During the last 12 months, 884/3774 (23.4%) of persons who had a fall were admitted to the hospital. Admittance to the hospital indicates a relatively severe injury in need of supplementary investigation, palliative treatment or observation but not necessarily a severe injury, like fracture or brain damage. Admission to the hospital after a fall was registered in NorCog and used as a marker of a fall injury. The reason for admission or the final diagnosis was not noted. Visits to emergency departments and hospital attendances have also been used as markers of serious injuries in other studies [2]. The observed incidence in this study (23.4%) is between the reported incidence of minor injuries (30–50%) and serious injuries (5–10%), which seems reasonable [3].

The bivariate analyses revealed several significant differences between fallers with and without an injury (i.e., hospital admission). Injuries were associated with female sex, older age, lower BMI, living alone, in need of public health service and walking assistance, reduced physical activity, reduced balance and gait speed, reduced activities of daily living, more comorbidity (cerebrovascular disease, COPD, and use of medication), and increased heart rate in standing position for 1 and 3 min. The high risk of injuries in persons living alone is most likely due to more risky activities when no help is available. In all, social factors, reduced somatic fitness and state of health characterized the injured. The increased heart rate in injured persons in the standing position for 1 and 3 min could indicate orthostasis. Also, the laboratory screen with low Hb and Ca and high homocysteine and ESR indicated increased morbidity, frailty, and malnutrition in the injured persons. The multivariable analysis confirms the finding of a reduced state of health in the injured persons. The reduced BMI could be an indicator of disease and malnutrition. Malnutrition is common in persons with dementia due to forgetfulness, and nutritional support reduces the risk of falls [27]. The state of nutrition in the study population is unknown. Low Ca could indicate osteoporosis, a known predictor of fractures. The need for public health services and walking assistance indoors and/or outdoors clearly indicates reduced physical performance and somatic health. Except for female sex and lower BMI, these predictors of injuries after falls were in accordance with those of falls reported in our previous study and other studies [3,4,7].

There were some significant differences between predictors of falls and fall-related injuries in the fallers. A low BMI was a predictor of fall-related injuries but not of falls, indicating a poor state of health and potential malnutrition in persons with fall-related injuries. Fall-related injuries occurred more often in females than men, possibly due to the higher prevalence of osteoporosis. In our previous study as well as other studies, falls were associated with cognitive impairment and type of dementia [3,4,7,28]. The fall risk increases between 2 and 20 times in persons with cognitive impairment [28]. In this study, neither the bivariate nor the multivariate analyses revealed any predictive effect of degrees of cognitive impairment (measured using the Cornell Depression Scale, Mayo Fluctuation Scale, MMSE-NR3, CERAD Delayed Recall and mild–moderate dementia) or dementia diagnoses on fall-related injuries. A report of a significantly higher incidence of fall-related injuries in persons with DLB than in those with AD could be due to the higher incidence of falls in persons with DLB and not a higher incidence of fall-related injuries in fallers with DLB [4,7,29]. If there were any differences in this study, the incidence of fall-related injuries was somewhat lower in persons with DLB and AD than in the other dementia groups. Only female sex, reduced general health (indicated by low BMI), reduced physical performance, and low Ca (indicating osteoporosis) were associated with fall-related injuries in the multivariable analyses. The marked difference between predictors of falls and fall-related injuries in fallers is the importance of impaired cognitive functions for falls and not for fall-related injuries. The difference is comprehensible. Persons with cognitive impairment or dementia might have reduced power of judgment and be inattentive, unconcerned and careless, which increases the fall incidence but not the risk of injury once falling. A reduced state of health with a high burden of comorbidity increases the risk of injury.

### Strengths and Limitations

The NorCog registry included consecutive persons attending a high number of Norwegian healthcare clinics dispersed throughout Norway for the evaluation of cognitive functions, which assures the representativeness of the population with cognitive impairment. NorCog included home-dwelling persons only; the inclusion of frail community-dwelling persons could have influenced the results. The testing was standardized, and the diagnoses were according to standard criteria. As an inclusion criterion was the ability to give informed consent for participation, only persons with a mild/moderate degree of cognitive impairment/dementia were included. The lack of reason for admission to the hospital is a weakness. The sample size was reduced to variable degrees in the analyses due to missing data. Therefore, the number of persons in each analysis is reported. The total sample size was judged as sufficient to give reliable results, as were the analyses with reduced samples. The statistically significant *p*-value was set to <0.01 due to the high number of analyses. Independent of the method used for adjusting for multiple testing, both type I and II errors will likely occur.

## 5. Conclusions

Falls are common in home-dwelling older persons with impaired cognitive function, and the degree of impairment is related to the fall incidence. In those with impaired cognitive function/dementia and falls, fall-related injuries were associated with reduced general health status and physical fitness and not with the degree of cognitive impairment and type of dementia. Prevention of fall-related injuries in persons at risk of falls should, therefore, focus on improved somatic health and physical performance rather than on relief of cognitive impairment. This strategy also reduces the overall risk of falls.

## Figures and Tables

**Table 1 geriatrics-10-00074-t001:** Characteristics of persons with and without a fall-related injury in the past 12 months with comparisons between the groups. The results are given as a number (proportion) or mean (SD).

Variable	Number	Fall w/Injury	Fall wo/Injury	Statistics *p*-Value
Gender: Male Female	1778 1996	366 (20.6%) 518 (26.0%)	1412 (79.4%) 1478 (74.0%)	***p* < 0.001**
Age (year)	3774	77.4 (8.2)	75.9 (8.6)	***p* < 0.001**
BMI (kg/m^2^)	3083	25.2 (4.5)	25.9 (4.8)	***p* = 0.002**
Education (year)	3497	10.5 (3.4)	10.8 (3.5)	*p* = 0.063
Living alone at home: Yes No	1554 2115	414 (26.6%) 438 (20.7%)	1140 (73.4%) 1677 (79.3%)	***p* < 0.001**
Need for public health service: Yes No	1750 1930	527 (30.1%) 340 (17.6%)	1223 (69.9%) 1590 (82.4%)	***p* < 0.001**
Walking assistance: No Indoor or outdoor Both in- and outdoor	1518 393 624	282 (18.6%) 104 (26.5%) 221 (35.4%)	1236 (81.4) 289 (73.5%) 403 (64.6%)	Trend: ***p* < 0.001**
Physical activity (score 0–8)	3226	2.48 (2.27)	2.87 (2.42)	***p* < 0.001**
SPPB Walking (order 0–4)	1625	2.82 (1.13)	3.13 (1.02)	***p* < 0.001**
SPPB Balance (seconds)	796	6.12 (5.43)	6.61 (6.18)	*p* = 0.378
PADL (score 6–30)	3595	9.70 (4.09)	8.59 (3.32)	***p* < 0.001**
IADL (score 8–31)	3652	17.40 (6.40)	15.96 (6.12)	***p* < 0.001**
Blood pressure (syst.) sitting (mmHg)	3214	143.4 (24.5)	144.4 (22.8)	*p* = 0.316
Blood pressure (diastolic) sitting	3213	80.9 (13.6)	81.3 (12.3)	*p* = 0.376
Heart rate sitting (beats/min)	3175	74.5 (15.0)	73.3 (13.6)	*p* = 0.036
Blood pressure (systolic) standing 1 min	2606	138.6 (25.5)	138.8 (24.8)	*p* = 0.864
Blood pressure (diastolic) standing 1 min	2603	80.5 (14.4)	80.6 (18.9)	*p* = 0.896
Heart rate standing 1 min	2552	80.5 (15.4)	78.5 (14.7)	***p* = 0.005**
Blood pressure (systolic) standing 3 min	2491	143.2 (25.9)	142.3 (24.5)	*p* = 0.482
Blood pressure (diastolic) standing 3 min	2487	82.5 (14.3)	82.2 (13.5)	*p* = 0.661
Heart rate standing 3 min	1902	82.1 (42.2)	78.5 (14.8)	***p* = 0.005**

SPPB: Short Physical Performance Battery. PADL: Personal Activities of Daily Living. IADL: Instrumental Activities of Daily Living. Statistically significant *p*-values are in bold.

**Table 2 geriatrics-10-00074-t002:** Medical history, cognitive functions, and drug treatment in persons with and without an injury after a fall within the last 12 months, with comparisons between the groups. The results are given as a number (proportion) or mean (SD).

Variable	Number	Fall w/Injury	Fall wo/Injury	Statistics *p*-Value
Cerebrovascular disease: Yes No	1007 2767	288 (28.6%) 596 (21.5%)	719 (71.4%) 2171 (78.5%)	***p* < 0.001**
Parkinson’s disease: Yes No	199 3575	36 (18.1%) 848 (23.7%)	163 (81.9%) 2727 (76.3%)	*p* = 0.071
Neurological disease: Yes No	711 3063	180 (25.3%) 704 (23.0%)	531 (74.7%) 2359 (77.0%)	*p* = 0.185
Coronary heart disease: Yes No	2264 1510	562 (24.8%) 322 (21.3%)	1702 (75.2%) 1188 (78.7%)	*p* = 0.013
Cardiac surgery: Yes No	171 3603	38 (22.2%) 846 (23.5%)	133 (77.8%) 2757 (76.5%)	*p* = 0.782
Diabetes: Yes No	606 3168	147 (24.3%) 737 (23.3%)	459 (75.7%) 2431 (76.7%)	*p* = 0.601
Cancer: Yes No	454 3320	85 (18.7%) 799 (24.1%)	369 (81.3%) 2521 (75.9%)	*p* = 0.011
COPD: Yes No	306 3468	92 (30.1%) 792 (22.8%)	214 (69.9%) 2676 (77.2%)	***p* = 0.006**
Polyarthritis: Yes No	210 3564	45 (21.4%) 839 (23.5%)	165 (78.6%) 2725 (76.5%)	*p* = 0.557
Depression (actual) Yes No	439 3335	102 (23.2%) 782 (23.4%)	337 (76.8%) 2553 (76.6%	*p* = 0.952
Cornell depression (scale 0–38)	2627	7.85 (5.88)	7.49 (5.96)	*p* = 0.200
MAYO Sleep (scale 0–10)	721	6.88 (2.38)	7.25 (2.35)	*p* = 0.075
MAYO Fluctuation (scale 0–4)	1439	1.62 (1.28)	1.59 (1.24)	*p* = 0.708
MAYO Fluctuation: Normal (score < 3) Abnormal (score ≥ 3)	1083 356	242 (22.3%) 86 (24.2%)	841 (77.7%) 270 (75.8%)	*p* = 0.512
Duration of cognitive impairment (years)	2345	3.16 (3.23)	3.33 (3.67)	*p* = 0.328
MMSE-NR3 (score 0–30)	2927	22.58 (4.65)	23.00 (4.61)	*p* = 0.038
CERAD–Immediate recall	3086	11.69 (4.86)	11.89 (4.94)	*p* = 0.345
CERAD–delayed recall	3043	2.21 (2.19)	2.30 (2.22)	*p* = 0.304
TMT-A (seconds)	3251	102.8 (64.7)	97.1 (64.6)	*p* = 0.033
TMT-B (seconds)	1563	214.3 (102.6)	202.9 (108.0)	*p* = 0.079
Drugs (number)	2511	5.67 (3.27)	5.04 (3.41)	***p* < 0.001**

COPD: Chronic Obstructive Pulmonary Disease. MMSE: Mini-Mental State Examination. CERAD: Consortium to Establish a Registry for Alzheimer’s Disease. TMT: Trail Making Test. Statistically significant *p*-values are in bold.

**Table 3 geriatrics-10-00074-t003:** Blood and cerebrospinal fluid analyses and genetic tests in persons with and without fall-related injuries within the last 12 months, with comparisons between the groups. The results are given as mean (SD) or number (proportion).

	Number	Fall w/Injury	Fall wo/Injury	Statistics *p*-Value
**Blood tests**				
Hb (g/100 mL)	2716	13.3 (1.8)	13.7 (1.7)	***p* < 0.001**
CRP (mg/L)	2461	5.9 (11.5)	5.3 (9.6)	*p* = 0.245
ESR (mm/1 h)	1510	17.0 (17.1)	14.0 (14.6)	***p* = 0.003**
Thrombocytes (10^9^/L)	2517	255 (88)	247 (74)	*p* = 0.036
Creatinine (µmol/L)	2744	87.8 (42.8)	86.3 (41.1)	*p* = 0.404
Albumin (g/L)	2513	40.8 (5.1)	41.8 (13.1)	*p* = 0.062
Folic acid (nmol/L)	2320	20.9 (14.9)	20.1 (14.5)	*p* = 0.259
Cholesterol (mmol/L)	2170	5.0 (2.2)	5.0 (1.8)	*p* = 0.512
Homocysteine (µmol/L)	1553	16.9 (6.9)	15.8 (6.6)	***p* = 0.005**
Na (mmol/L)	2706	140.2 (3.0)	140.6 (3.0)	*p* = 0.017
K (mmol/L)	2710	4.3 (0.5)	4.3 (0.4)	*p* = 0.853
Ca (mmol/L)	2146	2.23 (0.41)	2.29 (0.34)	***p* = 0.004**
Thyroxin (pmol/L)	2439	15.6 (3.1)	15.6 (4.2)	*p* = 0.901
TSH (mU/L)	2585	1.89 (1.23)	1.86 (1.20)	*p* = 0.559
Vitamin B12 (pmol/L)	2568	400 (282)	388 (270)	*p* = 0.317
Vitamin D (nmol/L)	665	74.3 (31.3)	71.9 (26.8)	*p* = 0.401
HbA1c (mmol/mol)	610	42.3 (13.2)	42.0 (11.4)	*p* = 0.776
Methyl Malonate (µmol/L)	878	0.25 (0.14)	0.24 (0.26)	*p* = 0.787
ALAT (U/L)	2664	21.5 (16.9)	21.9 (12.8)	*p* = 0.533
Gamma-GT (U/L)	1778	49.4 (77.1)	40.8 (48.8)	*p* = 0.033
ALP (U/L)	2339	79.7 (39.3)	76.8 (33.9)	*p* = 0.096
**Spinal fluid**				
Phospho-tau (pg/mL)	444	57.3 (33.7)	67.4 (43.5)	*p* = 0.047
Total-tau (pg/mL)	445	434 (299)	481 (300)	*p* = 0.098
Beta-amyloid (ng/L)	445	737 (320)	751 (329)	*p* = 0.742
**Genes**	545			
APOE E2E2	1	1 (100%)	0 (0%)	Chi-
APOE E2E3	33	5 (15%)	28 (85%)	square
APOE E2E4	18	4 (22%)	14 (78%)	Pearson
APOE E3E3	237	50 (21%)	187 (79%)	*p* = 0.390
APOE E3E4	204	38 (19%)	166 (81%)	Lin-by-lin *
APOE E4E4	52	12 (23%)	40 (77%)	*p* = 0.964

* Lin-by-Lin: Linear-by-linear is a chi-square test for trends. Statistically significant *p*-values are in bold.

**Table 4 geriatrics-10-00074-t004:** Comparisons of cognitive impairment and dementia diagnoses in persons with and without a fall-related injury.

Variable	Number	Fall w/Injury	Fall wo/Injury	Statistics *p*-Value
Clinical impairment -SCI -MCI -Dementia	3026 181 1144 1701	36 (19.9%) 269 (23.5%) 406 (23.9%)	145 (80.1%) 875 (76.5%) 1295 (76.1%)	Pearson *p* = 0.486 Lin-by-Lin * *p* = 0.365
Dementia diagnoses: AD AD/VaD VaD FTD DLB	537 240 126 115 12 44	47 (19.6) 39 31.0%) 31 (27.0%) 3 (25.0%) 8 (18.2%)	193 (80.4%) 87 (69.0%) 84 (73.0%) 9 (75.0%) 36 (81.8%)	Pearson *p* = 0.121

* Lin-by-Lin: Linear-by-linear is a chi-square test for trends.

**Table 5 geriatrics-10-00074-t005:** The statistically significant and most important (according to the Wald value) predictors of fall-related injuries in persons who have experienced a fall within the past 12 months. Logistic regression analysis with fall injury as the dependent variable. The number of persons in the analysis was 1288, of whom 319 (24.8%) had a fall injury.

Dependent Variables	B	Wald	OR	95% CI	*p*-Value
Sex (male)	−0.074	0.294	0.929	0.712; 1.212	*p* =0.588
Age (years)	−0.018	3.861	0.982	0.965; 1.000	*p* = 0.049
BMI (kg/m^2^)	−0.047	10.304	0.954	0.927; 0.982	***p* = 0.001**
Public health service	0.445	9.535	1.561	1.177; 2.071	***p* = 0.002**
Walking assistance	0.366	18.601	1.443	1.221; 1.702	***p* < 0.001**
Serum-Ca (mmol/L)	−0.450	6.764	0.637	0.454; 0.895	***p* = 0.009**

Hosmer–Lemeshow Goodness of Fit: Chi-square = 11.422, *p* = 0.179. Statistically significant *p*-values are in bold.

## Data Availability

The national registry NorCog is responsible for the source data. The de-identified data files from persons with falls were transferred to Innlandet Hospital Trust Brumunddal, Norway and stored on a server dedicated to research. The security follows the rules given by The Norwegian Data Protection Authority, P.O. Box 8177 Dep. NO-0034 Oslo, Norway. The data are available upon request to the author.

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
