# Peer review of "Predictors of Fall-Related Injuries in Fallers—A Study in Persons with Cognitive Impairment"

_geriatrics, 2025, doi:10.3390/geriatrics10030074_

Round 1
Reviewer 1 Report
Comments and Suggestions for Authors
Dear Authors,
Your paper is interesting, well-structured and addresses an important issue in public health. However, I think there are some aspects which can be more thoroughly described in your paper.
Table 1.
- Patients living alone had more falls with injury compared to those living with someone. How do you explain that? Normally they should be less frail, but receive less supervision in their daily activities. Did you make any difference between the elderly living with family members and those living in institutions?
- Please explain how you comment the higher heart rate in people with injury. It may be due to orthostasis, but in the analysis of blood pressure you did not observe any statistical differences.
Table 3: Please provide more context for the increased ESR in fallers with injury, as it usually predicts an increased frailty or risk for malnutrition.
Author Response
Comment to Table 1:
Patients living alone had more falls with injury compared to those living with someone. How do you explain that? Normally they should be less frail, but receive less supervision in their daily activities. Did you make any difference between the elderly living with family members and those living in institutions?
Answer:
The most likely reason for more falls with injuries in persons living alone is that they perform more risky activities when no help is available. This has been added on page 8, lines 213-214. Living alone means living alone at home. This has been clarified on page 2, line 78, and page 4, table 1. No difference was made between living at home and in institutions.
Comment:
Please explain how you comment the higher heart rate in people with injury. It may be due to orthostasis, but in the analysis of blood pressure you did not observe any statistical differences.
Answer:
A comment about the higher heart rate in the standing position has been added on page 8, lines 213 and 216-217.
Comment to Table 3:
Table 3: Please provide more context for the increased ESR in fallers with injury, as it usually predicts an increased frailty or risk for malnutrition.
Answer:
A comment has been added about the changes in the laboratory screen, including ESR, and the possibilities of increased frailty and malnutrition, page 8, lines 217-219.
Reviewer 2 Report
Comments and Suggestions for Authors
• A brief summary
This is a one-year follow-up of fallers among participants in the Norwegian Registry of Persons Assessed for Cognitive Symptoms study to explore predictors of fall-related injuries in fallers and compare these predictors with those of falls. The main contribution is an innovative idea to distinguish fall-injury predictors from fall predictors, and the finding regarding the role of cognition, which was not a predictor of fall-injury, is a surprising result. This is an interesting finding, which needs to be further investigated.
General concept comments
A major weakness is that predictors of falls and fall-injury cannot be compared in the method used because falls were predicted among fallers and non-fallers, but the fall-injury was predicted among fallers. The study consists of two independent purposes. A problem is that the fallers with and without injury were compared using ≤3774 participants, but the prediction used data of 1288 fallers, which were 34.1% of the fallers. The study deleted 63.9% of fallers for the analysis due to missing data.
Since no characteristic data of 1288 participants are present, we cannot understand who they are and if the data are representative, as the authors claimed. Especially, the number of fallers who had mild dementia and their cognitive status should be clarified.
For the logistic regression, was cognitive status entered to conclude that cognitive status was not a predictor of fall injury? Or is it because of the results of univariate analysis, for which it was not entered into the prediction analysis? Is this result due to similar mild cognitive statuses of many fallers?
If more than mild severity of cognitive decline was included, the result may vary. There are many incidents of injury due to falls from a bed and a wheelchair in older adults with dementia. Therefore, cognitive status may become a predictor of fall injury. To conclude, a statement, such as “in people with mild cognitive impairment,” should be present in the abstract, results, and conclusion.
• Specific comments
When many analyses are conducted, instead of using an arbitrary p<.01 (line 148), use a method to correct inflated Type I error rates, such as Bonferroni, Benjamini, Holm correction methods, etc.
Author Response
Comment: A brief summary
This is a one-year follow-up of fallers among participants in the Norwegian Registry of Persons Assessed for Cognitive Symptoms study to explore predictors of fall-related injuries in fallers and compare these predictors with those of falls. The main contribution is an innovative idea to distinguish fall-injury predictors from fall predictors, and the finding regarding the role of cognition, which was not a predictor of fall-injury, is a surprising result. This is an interesting finding, which needs to be further investigated.
Answer to “A brief summary”
We are glad to know that you appreciated our idea, and that the results are interesting and need further investigation despite some methodological limitations.
Comment:
General concept comments
A major weakness is that predictors of falls and fall-injury cannot be compared in the method used because falls were predicted among fallers and non-fallers, but the fall-injury was predicted among fallers. The study consists of two independent purposes. A problem is that the fallers with and without injury were compared using ≤3774 participants, but the prediction used data of 1288 fallers, which were 34.1% of the fallers. The study deleted 63.9% of fallers for the analysis due to missing data.
Answer to “General concept comments”:
It is correct that falls were predicted in the total population (i.e., persons with and without a fall), and fall injury was predicted in the population with a fall. The results cannot be compared directly, but these aims are not independent, since fall injuries occur only in persons who fall.
It is correct that 3774 persons with a fall last year were included in this study, and only 1288 (34.1%) were included in the analysis predicting independent variables of fall injuries (i.e. the multivariable analysis) due to missing data. The missing data was a significant study limitation that was clearly reported on page 9, lines 262-263. To present all results in an honest way and not hide the limitations of missing data, the number of persons in each of the analyses was reported. No changes have been made.
Comment:
Since no characteristic data of 1288 participants are present, we cannot understand who they are and if the data are representative, as the authors claimed. Especially, the number of fallers who had mild dementia and their cognitive status should be clarified.
Answer:
The characteristics of the 1288 persons in the multivariable analysis were not presented because it was considered unnecessary. Tables 1-4 give detailed information on all the variables. The multivariable analysis with 1288 persons (Table 5) is a summary of the detailed information in the tables. The reviewer’s specific question about the cognitive status of fallers with and without injuries is given in Table 4.
Comment:
For the logistic regression, was cognitive status entered to conclude that cognitive status was not a predictor of fall injury? Or is it because of the results of univariate analysis, for which it was not entered into the prediction analysis? Is this result due to similar mild cognitive statuses of many fallers?
Answer:
In addition to the six statistically significant variables in Table 5, all variables (including cognitive status) were added to the logistic regression model, one-by-one (stepwise). They were not selected based on the results of the univariate analyses. None of the other variables were statistically significant. The degree of cognitive impairment is given in Table 4. More than half of the participants were diagnosed with dementia, the most severe degree of cognitive impairment.
Comment:
If more than mild severity of cognitive decline was included, the result may vary. There are many incidents of injury due to falls from a bed and a wheelchair in older adults with dementia. Therefore, cognitive status may become a predictor of fall injury. To conclude, a statement, such as “in people with mild cognitive impairment,” should be present in the abstract, results, and conclusion.
Answer:
As can be seen in Table 4, the proportion of participants with subjective, mild and severe cognitive impairment was 6%, 38% and 56%, respectively. Thus, the majority had severe cognitive impairment, and the degree of cognitive impairment was not a predictor of fall injury. To include a statement such as “in people with mild cognitive impairment” will be erroneous.
Comment: Specific comments
When many analyses are conducted, instead of using an arbitrary p<.01 (line 148), use a method to correct inflated Type I error rates, such as Bonferroni, Benjamini, Holm correction methods, etc.
Answer:
Adjusting for multiple testing is a recurring discussion. Methods like Bonferroni, Benjamini-Hochberg (False Discovery Rate) etc. are well known, but there is no agreement on when and how to use them. This study reports 77 p-values. When correcting for multiple testing, Is it correct to include all p-values at the same time, or to include the p-values groupwise (e.g. one group could be one table)? The results will differ. Independent of the method used for adjusting for multiple testing, type I and II errors are inevitable. Adjusting the significant p-value to < 0.01 is a pragmatic and generally accepted way to do the adjustment. A sentence has been added, page 9, lines 265-267.
Round 2
Reviewer 2 Report
Comments and Suggestions for Authors
This is a revised version. There are still concerns.
- Although the study found an interesting result: cognition does not affect fall injury, the conclusion and abstract need to be clearer about the sample. The sample excludes older adults with more than moderate or severe dementia. If the sample consists of community-dwelling older adults, it also should be mentioned. We know nursing home residents fall often, and their fall injury tends to be severe. Therefore, without this clarification, the results would not be trustworthy.
- The authors say that the characteristics of 1288 people used to find significant variables associated with fall injury are not important, but they are important. We need to know who they are. What percentages of 1288 had various cognitive levels, illnesses, age, sex, living status, etc.? Were there any differences between the two groups in this particular sample? They represent only 34.1% of 3744, therefore, they must vary from those of 3774. If the space is limited, the selected characteristics can be reported.
- Report of univariate analyses of 77(?) variables is good, but they are independent of the multivariate prediction. If both analyses are important, these two aims should be mentioned before the method section. If the second aim is the focus, the univariate analyses for 1288 should be included.
- Unless there is a predetermined p-value for hypothesis testing, use a correction method and state the weakness of the approach. Bonferroni or Benjamini is not the choice; FDR can be used. The results for the 3774 may change.
Author Response
Answer to Reviewer 2
This is a revised version. There are still concerns.
1. Although the study found an interesting result: cognition does not affect fall injury, the conclusion and abstract need to be clearer about the sample. The sample excludes older adults with more than moderate or severe dementia. If the sample consists of community-dwelling older adults, it also should be mentioned. We know nursing home residents fall often, and their fall injury tends to be severe. Therefore, without this clarification, the results would not be trustworthy.
Answer:
It is not correct that the sample excludes older adults with more than moderate or severe dementia. Table 4 gives the cognitive status of the fallers with and without injuries. All participants lived at home (not in nursing homes). This has been added on page 2, lines 66-67.
2. The authors say that the characteristics of 1288 people used to find significant variables associated with fall injury are not important, but they are important. We need to know who they are. What percentages of 1288 had various cognitive levels, illnesses, age, sex, living status, etc.? Were there any differences between the two groups in this particular sample? They represent only 34.1% of 3744, therefore, they must vary from those of 3774. If the space is limited, the selected characteristics can be reported.
Answer:
This comment was answered in our previous report to the reviewer.
3. Report of univariate analyses of 77(?) variables is good, but they are independent of the multivariate prediction. If both analyses are important, these two aims should be mentioned before the method section. If the second aim is the focus, the univariate analyses for 1288 should be included.
Answer:
The sample sizes of the univariate and multivariate analyses differ because of missing values, but the samples are not independent. The sample used for the multivariable analysis is a part of the samples used for the univariate analyses. The aims of the univariate and multivariate analyses are the same. The results give the unadjusted and adjusted predictors of fall injuries.
4. Unless there is a predetermined p-value for hypothesis testing, use a correction method and state the weakness of the approach. Bonferroni or Benjamini is not the choice; FDR can be used. The results for the 3774 may change.
Answer:
This comment from the reviewer was responded to in our previous report.